# Endothelial Dysfunction Is Associated with Decreased Nitric Oxide Bioavailability in Dysglycaemic Subjects and First-Degree Relatives of Type 2 Diabetic Patients

**DOI:** 10.3390/jcm11123299

**Published:** 2022-06-09

**Authors:** Ignatios Ikonomidis, George Pavlidis, Maria Tsoumani, Foteini Kousathana, Konstantinos Katogiannis, Damianos Tsilivarakis, John Thymis, Aikaterini Kountouri, Emmanouil Korakas, Loukia Pliouta, Athanasios Raptis, John Parissis, Ioanna Andreadou, Vaia Lambadiari

**Affiliations:** 12nd Department of Cardiology, Attikon University Hospital, Medical School, National and Kapodistrian University of Athens, 12462 Athens, Greece; geo_pavlidis@yahoo.gr (G.P.); kenndj89@gmail.com (K.K.); tsilyd@yahoo.com (D.T.); johnythg@gmail.com (J.T.); jparissis@yahoo.com (J.P.); 2Laboratory of Pharmacology, Faculty of Pharmacy, National and Kapodistrian University of Athens, 15741 Athens, Greece; marietsoumani@gmail.com (M.T.); jandread@pharm.uoa.gr (I.A.); 32nd Department of Internal Medicine, Research Unit and Diabetes Center, Medical School, National and Kapodistrian University of Athens, 12462 Athens, Greece; f.kousathana@hotmail.com (F.K.); katerinak90@hotmail.com (A.K.); mankor-th@hotmail.com (E.K.); plioutaloukia@gmail.com (L.P.); atraptis@med.uoa.gr (A.R.); vlambad@otenet.gr (V.L.)

**Keywords:** oxidative stress, nitric oxide, endothelial function, diabetes, insulin resistance, first-degree relatives

## Abstract

Oxidative stress plays an important role in the pathogenesis of diabetes. We investigated oxidative stress and nitrite/nitrate concentrations at baseline and during postprandial hyperglycaemia in 40 first-degree relatives (FDRs) of diabetic patients with normal oral glucose tolerance test (OGTT) results, 40 subjects with abnormal OGTT results (dysglycaemic) and 20 subjects with normal OGTT results (normoglycaemic). Malondialdehyde (MDA), protein carbonyls (PCs), nitrite/nitrate plasma levels, the perfused boundary region (PBR—Glycocheck) of the sublingual microvessels, a marker of glycocalyx integrity, coronary flow reserve (CFR) and left ventricular global longitudinal strain (GLS) were assessed at 0 and 120 min of the OGTT. Insulin sensitivity was evaluated using Matsuda and the insulin sensitivity index (ISI). In all subjects, there were no significant changes in MDA or PC after the OGTT (*p* > 0.05). Compared with normoglycaemic subjects, FDRs and dysglycaemic subjects had significantly decreased nitrite/nitrate levels (−3% vs. −24% vs. −30%, respectively), an increased PBR and reduced CFR and GLS at 120 min (*p* < 0.05). The percent reduction in nitrite/nitrate was associated with abnormal Matsuda and ISI results, reversely related with the percent increase in PBR (r = −0.60) and positively related with the percent decrease in CFR (r = 0.39) and GLS (r = 0.48) (*p* < 0.05). Insulin resistance is associated with reduced nitric oxide bioavailability and coronary and myocardial dysfunction in FDRs and dysglycaemic subjects.

## 1. Introduction

Oxidative stress plays an important role in the pathogenesis of type 2 diabetes mellitus (T2DM) and related cardiovascular complications by promoting atherosclerotic process [1,2,3]. Previous studies have demonstrated the interaction between lipid peroxidation and protein oxidation, as assessed by malondialdehyde and protein carbonyls, respectively, and endothelial, arterial and left ventricular (LV) myocardial dysfunction in patients with T2DM [4,5]. The reduction in nitric oxide (NO) bioavailability is a hallmark of hyperglycaemia-induced endothelial dysfunction [6]. Plasma nitrite and nitrate concentrations, as oxidative products of NO metabolism, reflect the acute changes in endothelial NO synthase (eNOS) activity [7].

Endothelial glycocalyx is a layer composed of glycoproteins, proteoglycans and glycosaminoglycans that prevent direct contact of circulating blood cells with the endothelium surface. Damage to glycocalyx integrity has been shown to occur during acute and long-term hyperglycaemia, leading to the promotion of early atherogenic processes [5]. Hence, glycocalyx degradation is an independent predictor of adverse outcomes in subjects without clinically overt cardiovascular disease [8]. A non-invasive imaging technique has been developed to estimate endothelial glycocalyx of the sublingual microvessels using a dedicated camera [9]. Furthermore, our previous studies have shown that coronary flow reserve (CFR), a marker of coronary microcirculation, as well as LV myocardial strain, is impaired in first-degree relatives (FDR) of diabetic patients and dysglycaemic subjects compared with normoglycaemic subjects [10,11]. A recent study revealed that subjects with normal glucose tolerance with one-hour glucose levels ≥ 155 mg/dL during an oral glucose tolerance test (OGTT) had subclinical impairment in LV myocardial performance, and this finding was correlated with increased oxidative stress [12]. However, little is known about the association of endothelial glycocalyx, coronary microcirculation and LV myocardial function with the kinetics of oxidative stress markers and nitrite/nitrate levels during postprandial hyperglycaemia.

Thus, the aim of the present study was to investigate (a) the differences in baseline oxidative stress markers, nitrite/nitrate plasma levels, endothelial glycocalyx integrity, coronary microcirculation and LV myocardial strain between FDR, dysglycaemic and normoglycaemic subjects and (b) the acute changes in oxidative stress markers and nitrite/nitrate plasma levels in relation to the corresponding changes in vascular and cardiac indexes after postprandial hyperglycaemia provoked during a 2 h OGTT.

## 2. Materials and Methods

### 2.1. Study Design

One hundred ten subjects who were referred to our hospital outpatient clinics were assessed for eligibility by the attending internists (G.P., F.K. and V.L.). Inclusion criteria were male or female patients aged 18 to 70 years without known diabetes and with or without parental history of diabetes. Exclusion criteria were history of coronary artery disease, moderate or severe valvular disease, heart failure, peripheral vascular disease, malignant neoplasms, chronic kidney disease, liver insufficiency and history of alcohol abuse. All women who participated in the present study were premenopausal, and none of them were taking contraceptives. Of the 110 enrolled subjects, 10 patients were excluded from the study because of unwillingness to participate (*n* = 6), history of alcohol abuse (*n* = 1) or inadequate echocardiography images for analysis (*n* = 3). Hence, 100 patients were included in the study (Figure 1).

We compared baseline oxidative stress markers, nitrite/nitrate levels, endothelial glycocalyx integrity, coronary flow reserve and LV myocardial strain in forty FDRs against forty dysglycaemic subjects and twenty normoglycaemic subjects. All participants underwent a 2 h OGTT, and the examined biochemical, vascular and cardiac markers were estimated at baseline and at 120 min of OGTT to evaluate the impact of postprandial hyperglycaemia.

### 2.2. Study Population

We examined three study groups. The first group consisted of forty first-degree relatives of type 2 diabetic patients with at least one parent with diabetes mellitus and who completed a 75-g OGTT with normal results (plasma glucose levels < 140 mg/dL at the 2 h mark of the OGTT) [13]. First-degree relatives were recruited through their parents, who attended the outpatient diabetic clinic. According to the current guidelines, it is recommended to perform a screening test for prediabetes and diabetes, including an OGTT every 3 years in FDRs [13]. The second study group consisted of forty subjects matched for age and sex with an FDR, with abnormal OGTT results (dysglycaemic subjects; plasma glucose levels > 140 mg/dL at the 2 h sample of the OGTT). In these subjects, OGTT was performed as a screening test for diabetes prevention due to their family history of diabetes. The third group consisted of twenty subjects, matched for age and sex with an FDR, without a parental history of diabetes, a fasting glucose concentration < 100 mg/dL, no use of antidiabetic agents and plasma glucose levels < 140 mg/dL at the 2 h OGTT sample (normoglycaemic subjects). In this group, OGTT was performed to exclude prediabetes or insulin resistance. In the present study, the fasting glucose concentrations and the glucose values at the 2 h OGTT sample were within the normal range in normoglycaemic subjects and FDRs [13]. In contrast, all dysglycaemic subjects (*n* = 40) showed glucose levels > 140 mg/dL at the 2 h sample of the OGTT. Of those, 23 subjects had glucose levels at 120 min between 140 and 199 mg/dL and were defined as having impaired glucose tolerance (prediabetic subjects), whereas the remaining 17 subjects showed glucose levels at 120 min sample of the OGTT > 200 mg/dL and were defined as diabetic patients [13]. Propensity score analysis was performed to ensure that the three study groups were adequately balanced for the traditional cardiovascular disease risk factors, body mass index (BMI) and waist and hip circumferences.

### 2.3. Primary and Secondary Endpoints

The primary outcome was the differences in baseline oxidative stress markers, nitrite/nitrate plasma levels, endothelial glycocalyx integrity, coronary microcirculation and LV myocardial strain between FDR, dysglycaemic and normoglycaemic subjects. Secondary outcomes were acute changes in oxidative stress markers and nitrite/nitrate plasma levels in relation to the corresponding changes in vascular and cardiac indexes after postprandial hyperglycaemia provoked during a 2 h OGTT in the three study groups.

### 2.4. Laboratory Measurements

In all participants, a standard 75-g OGTT was performed. Venous blood samples were drawn at 0, 30, 60, 90 and 120 min after oral glucose loading to measure plasma glucose and insulin concentrations. Glucose tolerance status was determined on the basis of OGTT as described by the World Health Organization [14]. Normal glucose tolerance was considered a 2 h post-loading-glucose value lower than 140 mg/dL, and impaired glucose tolerance was considered a 2 h plasma glucose value between 140 and 199 mg/dL. A 2 h post-loading-glucose value of 200 mg/dL or higher was diagnostic of type 2 diabetes mellitus [13]. Insulin resistance during OGTT was determined using Matsuda index and insulin sensitivity index (ISI). Matsuda index is strongly associated with euglycaemic insulin clamp [15] and calculated using the formula: 10,000/square root of (fasting glucose × fasting insulin) × (mean glucose × mean insulin during OGTT). Insulin sensitivity index (ISI) represents a quantitative method for estimating the insulin resistance [16] and was calculated as follows: ISI_0,120_ = 75,000 + (fasting glucose − glucose at 120 min) × 0.19 × weight (kg)/120 × mean glucose × log(mean insulin).

Malondialdehyde and protein carbonyls were estimated spectrophotometrically using a commercial kit (Oxford Biomedical Research, Rochester Hills, MI, USA) with a colorimetric assay for lipid peroxidation (measurement range 1–20 nmol/L) and with the assessment of the 2,4-dinitrophenylhydrazine derivatives of protein carbonyls (nmol/mL), respectively, according to a previously published methodology [5]. Plasma nitrites and nitrates were determined using a spectrophotometric method (Nitrate/Nitrite Colorimetric Assay Kit 780001, Cayman Chemicals, Ann Arbor, MI, USA) with a previously described methodology [17].

### 2.5. Endothelial Glycocalyx Assessment

We assessed the endothelial glycocalyx of the sublingual microvessels non-invasively using a Sidestream Darkfield (SDF) camera (Microscan, GlycoCheck, Microvascular Health Solutions Inc., Salt Lake City, UT, USA) [11]. This camera is inserted under the tongue and provides a recording of greater than 3000 microvessels segments with a lumen diameter ranging from 5 to 25 μm. The images are analyzed via a dedicated software which detects the lateral erythrocytes’ movement into the glycocalyx luminal surface, which is reported as the perfused boundary region (PBR). The calculated PBR is presented as the mean of the sublingual microvessels with a diameter range of 5–25 μm and by a group of microvessels with diameters of 5–9 μm, 10–19 μm and 20–25 μm. Increased PBR values indicate a damaged glycocalyx that is more accessible for circulating red blood cells. The estimation of PBR utilizing SDF camera has satisfactory reproducibility and is operator-independent [8]. Additionally, this method was proposed as a validated technique for the assessment of endothelial function by the European Society of Cardiology Working Group of Peripheral Circulation [9]. The inter- and intra-observer variability in PBR assessment was 5.2% and 4.3%, respectively.

### 2.6. Coronary Flow Reserve Assessment

Coronary flow velocities in the left anterior descending (LAD) coronary artery were obtained using colour-guided pulsed-wave Doppler from long-axis apical view with a 7 MHz transducer. We measured the maximal diastolic velocity of the LAD coronary flow wave (CF-max) at baseline and during hyperaemia after adenosine infusion (140 µg/kg/min) for 3 min. CFR was calculated as the ratio of hyperaemic to resting CF-max [10]. Measurements of three cardiac cycles were averaged. The inter- and intra-observer variability of CFR assessment was 5% and 2%, respectively.

### 2.7. Echocardiography

Transthoracic echocardiography was performed in all participants using a Vivid Ε95 (GE Medical Systems, Horten, Norway) ultrasound system. All studies were digitally stored in a computerized station (EchoPac GE 203, Horten, Norway) and were analyzed by two observers blinded to the clinical and laboratory data.

### 2.8. Two-Dimensional Strain Analysis

Two-dimensional strain was estimated using speckle tracking analysis (EchoPac PC 203, GE Healthcare, Horten, Norway). We measured the LV global longitudinal strain (GLS, %) utilizing the 17 LV segment model imaged from the apical 4-, 2- and 3- chamber views (frame rate: 70–80/s), as previously mentioned in [11]. The inter- and intra-observer variability in GLS was 7% and 10%, respectively.

### 2.9. Statistical Analysis

We planned to investigate the percent change (Δ) of PBR from independent control (normoglycaemic subjects) and experimental subjects (first-degree relatives) with 0.5 control per experimental subject. Hence, we performed a pilot study that included 5 normoglycaemic subjects and 10 FDRs, and the response within each study group was normally distributed with a standard deviation equal to 0.3. The true difference between normoglycaemic subjects and FDRs in the mean values of PBR was 10%. Thus, we would need to study 20 normoglycaemic subjects and 40 FDRs to be able to reject the null hypothesis that the study population mean values for ΔPBR of normoglycaemic subjects and FDRs are equal with a probability (power) of 0.8 and a type 1 error probability of 0.05. To investigate whether subjects included in the three study groups (normoglycaemic subjects, FDRs and dysglycaemic subjects) were adequately balanced for the presence of atherosclerotic disease, we estimated the logit propensity score in each subject using a logistic regression model that included as dependent variables age, sex, dyslipidaemia, smoking, BMI, waist and hip circumferences and systolic and diastolic blood pressure as atherosclerotic factors. Subsequently, we compared propensity scores between the three study groups using one-way analysis of variance (ANOVA). All statistical calculations were carried out with the Statistical Package for Social Sciences version 25.0 for Windows (IBM SPSS Statistics, Inc., Chicago, IL, USA). Data are expressed as mean ± standard deviation. Categorical variables are expressed as percentages of the study population and were compared using the χ2 test. Continuous variables were compared using t-test. Simple correlations between the continuous variables were detected using parametric or non-parametric correlation coefficients. We performed full factorial ANOVA to examine the differences between the three study groups and ANOVA for repeated measurements to compare the examined markers at 0 and 120 min of the OGTT, which used a within-subject factor for each group separately in a model including age, sex, dyslipidaemia, smoking, BMI, waist and hip circumferences and systolic and diastolic blood pressure as covariates. Values of *p* lower than 0.05 were considered significant.

## 3. Results

### 3.1. Clinical and Biochemical Characteristics of the Three Study Groups

Table 1 shows the clinical and metabolic characteristics of normoglycaemic subjects, FDRs and dysglycaemic subjects. There were no significant differences among the study groups in age, sex or traditional cardiovascular risk factors (*p* > 0.05). The estimated propensity scores by logistic regression analysis, including age, sex, dyslipidaemia smoking, BMI, waist and hip circumferences and systolic and diastolic blood pressure, were similar among the three study groups (*p* = 0.764). Hence, the three study groups were adequately balanced for atherosclerotic markers.

First-degree relatives and dysglycaemic subjects had higher total cholesterol, low-density lipoprotein (LDL) cholesterol, triglycerides and fasting insulin values and lower high-density lipoprotein (HDL) cholesterol, Matsuda index and ISI scores than normoglycaemic subjects (*p* < 0.05; Table 1). First-degree relatives and dysglycaemic subjects had similar insulin values and Matsuda index scores (*p* > 0.05; Table 1). At 120 min after the glucose challenge, FDRs and dysglycaemic subjects had higher glucose and insulin levels compared to normoglycaemic subjects after adjustment for age, sex, dyslipidaemia, smoking, BMI, waist and hip circumferences and systolic and diastolic blood pressure (*p* < 0.05; Table 1).

### 3.2. Oxidative Stress Markers and Nitrite/Nitrate Levels at Baseline and at 120 min after Glucose Loading

In the fasting state, the plasma concentrations of malondialdehyde and protein carbonyls, as well as the levels of nitrites/nitrates, were similar among the three study groups after adjustment for age, sex, dyslipidaemia, smoking, BMI, waist and hip circumferences and systolic and diastolic blood pressure (*p* > 0.05; Table 2).

After the oral glucose loading, there were no significant changes in malondialdehyde or protein carbonyls in the three study groups (*p* > 0.05; Table 2; Figure 2). In FDRs and dysglycaemic subjects, nitrites, nitrates and their sum (nitrites + nitrates) were significantly decreased at 120 min, whereas they were remained unchanged in normoglycaemic subjects (−6% vs. −23% vs. −37%, *p* = 0.004, −3% vs. −24% vs. −29%, *p* = 0.002 and −3% vs. −24% vs. −30%, *p* < 0.001, respectively; Table 2; Figure 2).

In the group of dysglycaemic subjects, there were no significant differences in malondialdehyde, protein carbonyls or nitrite/nitrate levels between prediabetic (*n* = 23) and diabetic subjects (*n* = 17) at baseline (*p* > 0.05, data not shown). However, diabetic patients presented a greater reduction in nitrites, nitrates and in the sum of nitrites + nitrates (−42% vs. −35%, *p* = 0.038, −32% vs. −27%, *p* = 0.042, −34% vs. −28%, *p* = 0.035, respectively) compared with prediabetic subjects, whereas no significant changes were observed in malondialdehyde or protein carbonyls in either group after the OGTT.

### 3.3. Vascular and Cardiac Markers at Baseline and at 120 min after Glucose Loading

Compared to normoglycaemic subjects, dysglycaemic subjects and FDRs had higher PBR, indicating an impaired endothelial glycocalyx barrier function, after adjustment for age, sex, dyslipidaemia, smoking, BMI, waist and hip circumferences and systolic and diastolic blood pressure (*p* < 0.05; Table 2). On the other hand, dysglycaemic subjects had impaired coronary microcirculation, as estimated by reduced CFR, compared to normoglycaemic subjects and FDRs. Moreover, FDRs and dysglycaemic subjects showed lower values (less negative) of LV global longitudinal strain than normoglycaemic subjects (*p* < 0.05; Table 2).

Compared to normoglycaemic subjects, FDRs and dysglycaemic subjects showed an increase in PBR (−3% vs. +7% vs. +10%, respectively, *p* = 0.042) and a greater reduction in CFR (−6% vs. −11% vs. −14%, respectively, *p* = 0.033) and GLS (+0.1 vs. −4% vs. −4%, respectively, *p* = 0.021) at 120 min after OGTT (Table 2).

Among dysglycaemic subjects (*n* = 40), diabetic patients (*n* = 17; 120 min glucose levels > 200 mg/dL) had similar values of LV GLS, higher values of PBR (2.78 ± 0.5 μm vs. 2.42 ± 0.6 μm, *p* = 0.025) and more impaired CFR (2.66 ± 0.31 vs. 3.02 ± 0.38, *p* = 0.043) compared with prediabetic subjects (*n* = 23; 120 min glycose levels between 140 and 199 mg/dL) at baseline. After the OGTT, diabetic patients showed a greater increase in PBR (+13% vs. +8%, *p* = 0.012) and a greater reduction in CFR (−18% vs. −12%, *p* = 0.039), whereas the change in LV GLS at 120 min did not differ between the two groups (diabetic and prediabetic subjects; *p* > 0.05).

### 3.4. Interrelation between Changes in Nitrite/Nitrate Levels and Changes in Vascular Markers

In FDRs and dysglycaemic subjects, the percent reduction in nitrite/nitrate plasma concentrations at 120 min of OGTT was associated with the percentage increase in glucose at 60 min and 120 min (r = −0.45, *p* = 0.012 and r = −0.32, *p* = 0.026, respectively), insulin at 60 min and 120 min (r = −0.24, *p* = 0.045 and r = −0.38, *p* = 0.016, respectively) and with a decreased Matsuda (r = 0.37, *p* = 0.033) and ISI (r = 0.26, *p* = 0.046). In addition, the reduced nitrite/nitrate levels were reversely related with the impairment of endothelial glycocalyx barrier function during postprandial hyperglycaemia, as assessed by the percent increase in PBR (r = −0.60, *p* = 0.001) and positively related with the impairment of coronary microcirculation, as estimated by the percent decrease in CFR (r = 0.39, *p* = 0.038) and in LV myocardial deformation, as assessed by the percent decrease in LV global longitudinal strain (r = 0.48, *p* = 0.008).

## 4. Discussion

In the current study, we described oxidative stress and nitrite/nitrate status and their relationship with endothelial and myocardial function after a 2 h standard 75-g OGTT in FDRs of diabetic patients and dysglycaemic subjects compared with normoglycaemic subjects. The results of the present study show the following new findings: (1) there were no significant changes in malondialdehyde and protein carbonyls after OGTT in the three study groups (normoglycaemic subjects, FDRs and dysglycaemic subjects); (2) in FDRs and dysglycaemic subjects, nitrite/nitrate levels were significantly decreased at 120 min after the glucose challenge, whereas they remained unchanged in normoglycaemic subjects, and (3) the acute reduction in nitrite/nitrate concentrations was associated with decreased insulin sensitivity, as estimated by the Matsuda and ISI, as well as with impaired endothelial glycocalyx integrity, coronary flow reserve and LV myocardial function in FDRs and dysglycaemic subjects. Thus, insulin resistance and increased glucose levels combined with reduced nitrite/nitrate concentrations appear to be associated with acute endothelial, vascular and myocardial responses during postprandial hyperglycaemia.

In the present study, we demonstrated no significant differences in oxidative stress markers, namely malondialdehyde and protein carbonyls, in the three study groups after the OGTT. A previous study has shown that a transient, acute increase in plasma glucose levels after a standard OGTT per se does not cause a significant increase in markers of oxidative stress or a significant reduction in plasma antioxidants in non-diabetic subjects [18]. However, Yadav et al. have reported abnormal antioxidant levels in the first-degree relatives of type 2 diabetes mainly in subjects with low tolerance and intolerance to glucose [19], whereas Konukoğlu et al. showed that even mild hyperglycaemia can decrease erythrocyte glutathione levels, reflecting the presence of glucose-induced oxidative stress [20]. This compensatory reduction in antioxidant levels in response to increased post-glucose-loading oxidative stress may explain the non-significant changes in oxidative stress markers observed at 120 min of the OGTT in the present study. On the contrary, after the glucose tolerance test, nitrite/nitrate levels were reduced in FDRs and the dysglycaemic group. This finding is consistent with the study by Derosa et al. [21]. In this study, participants included 256 overweight healthy subjects and 274 overweight subjects with type 2 diabetes who underwent a 75-g OGTT. The results show that there was a significant reduction in nitrite/nitrate levels after the OGTT. Nitrites/nitrates reflect endothelial NO, and their reduction is an indirect marker of endothelial stress and a potential cause of an NO generation decrease from the endothelium [22]. Nitric oxide is synthesized from the amino acid L-arginine by eNOS [23]. Mah et al. showed that during OGTT, L-arginine is decreased with a simultaneous increase in the competitive inhibitor asymmetric dimethylarginine, leading to an overall decline in NO bioavailability [24]. Additionally, first-degree relatives of diabetic patients and dysglycaemic subjects, probably due to insulin resistance, cannot compensate for postprandial hyperglycaemia normally. Hence, high concentrations of superoxide anions, resulting from hyperglycaemia, rapidly react with NO to generate the strong oxidant peroxynitrate. Indeed, in our study, the acute reduction in nitrite/nitrate levels was correlated with decreased insulin sensitivity, as estimated by Matsuda and ISI.

We have previously demonstrated that FDRs have impaired endothelial glycocalyx integrity, coronary flow reserve and LV myocardial deformation compared to normoglycaemic subjects and similar endothelial glycocalyx and coronary microcirculatory dysfunction to that observed in dysglycaemic subjects [10,11]. Moreover, we have showed that FDRs present deteriorated endothelial glycocalyx, coronary flow reserve and LV myocardial function after postprandial hyperglycaemia in a similar pattern to dysglycaemic subjects, probably due to the similar level of insulin resistance that was observed in both FDRs and dysglycaemic individuals. In accordance with the abovementioned findings, in this study, we showed that an acute reduction in nitrite/nitrate concentrations was associated with impaired endothelial glycocalyx integrity, coronary microcirculatory function and LV myocardial function in FDRs and dysglycaemic subjects. Of note, the age of the subjects and the sample size of the present study are similar to those of the aforementioned studies. A disturbance in NO bioavailability is considered to be responsible for the functional changes that are related to endothelial dysfunction and cardiac damage, contributing to the development of atherosclerotic cardiovascular disease [23]. In vitro studies indicated that the exposure of endothelial cells to increased glucose concentrations leads to the formation of oxygen-derived free radicals, including superoxide anion, resulting in NO inactivation or contributing to functional damage to endothelial cells [25,26]. Endothelial dysfunction seems to occur even during relatively moderate hyperglycaemia, similar to that found in subjects with impaired fasting glucose or impaired glucose tolerance, whereas an increased glucose level rapidly induces endothelial dysfunction in healthy subjects with a family history of diabetes [27,28]. On the other hand, several studies have revealed that plasma levels of inflammatory mediators, such as interleukin (IL)-6, IL-8 and monocyte chemoattractant protine-3, acutely increased during OGTT [29,30]. Intriguingly, Esposito et al. showed that tumour necrosis factor-α levels are elevated in individuals with insulin resistance, and a high-fat meal induces a further increase in its circulating levels in association with reduced endothelial function as assessed by the L-arginine test [31]. Inflammation-mediated reduction in NO bioavailability may contribute to systemic microvascular dysfunction, as estimated in our study by impaired endothelial glycocalyx after the OGTT in FDRs and dysglycaemic subjects. In contrast, in a recent study, we demonstrated that IL-6 inhibition by tocilizumab improves endothelial glycocalyx, likely via a profound reduction in inflammatory burden and oxidative stress after a 3-month treatment [32]. Interestingly, accumulating preclinical and clinical evidence support the potential endothelial protective role of anti-hyperglycaemic agents such as metformin as well as gliflozins, which are now used in non-diabetic patients, by attenuating oxidative stress and endothelial inflammation [33,34].

However, Takei et al. reported that insulin resistance rather than oxidative stress and sympathetic activity, as assessed using plasma catecholamines, was associated with the reduction in coronary flow reserve after acute hyperglycaemia, probably due to increased myocardial oxygen demands and declined reactivity of vasodilation in the early stages of impaired glucose metabolism [35]. Indeed, in the current study, CFR was reduced significantly in FDRs and dysglycaemic subjects, likely on the grounds of insulin resistance, despite the fact that oxidative stress markers remained unchanged after the glucose challenge. Interestingly, CFR is linked to LV myocardial function due to its effect on the perfusion of subendocardial contractile fibres [11]. Thus, an impaired LV myocardial performance was observed after acute postprandial hyperglycaemia in FDRs and dysglycaemic but not in normoglycaemic subjects.

### Study Limitations

The limitation of the present study is the cross-sectional design, which did not allow us to investigate cause–effect relationships. In addition, the study population was relatively young, with a mean age of 43 years. This, together with the small sample size of the study, prevents the results from being extended to an older population. Further large-scale studies are required to expand our findings.

## 5. Conclusions

Insulin resistance is associated with acute endothelial responses during postprandial hyperglycaemia, leading to reduced nitric oxide production with resultant coronary and myocardial dysfunction in FDRs of diabetic patients and dysglycaemic subjects. It is the first time, to the best of our knowledge, that decreased nitric oxide bioavailability has been linked to impaired endothelial glycocalyx integrity, coronary microcirculation and left ventricular myocardial function. These findings suggest that prolonged and repeated postprandial hyperglycaemia may play a significant role in the development of the atherosclerotic process. The appropriate intervention for insulin resistance, even in the prediabetes stage, and the prompt treatment of coexisting atherosclerotic risk factors, may effectively reduce the occurrence of diabetic heart disease.

## Figures and Tables

**Figure 1 jcm-11-03299-f001:**
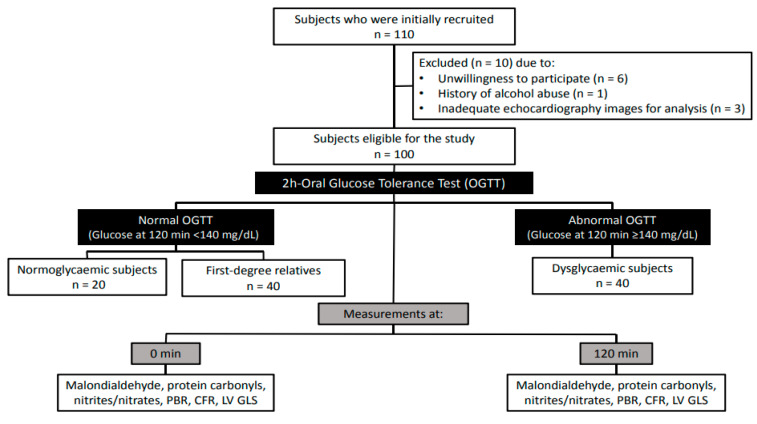
Flowchart depicting the study design including time points at which measurements of the examined markers were performed. PBR, perfused boundary region; CFR, coronary flow reserve; LV GLS, left ventricular global longitudinal strain.

**Figure 2 jcm-11-03299-f002:**
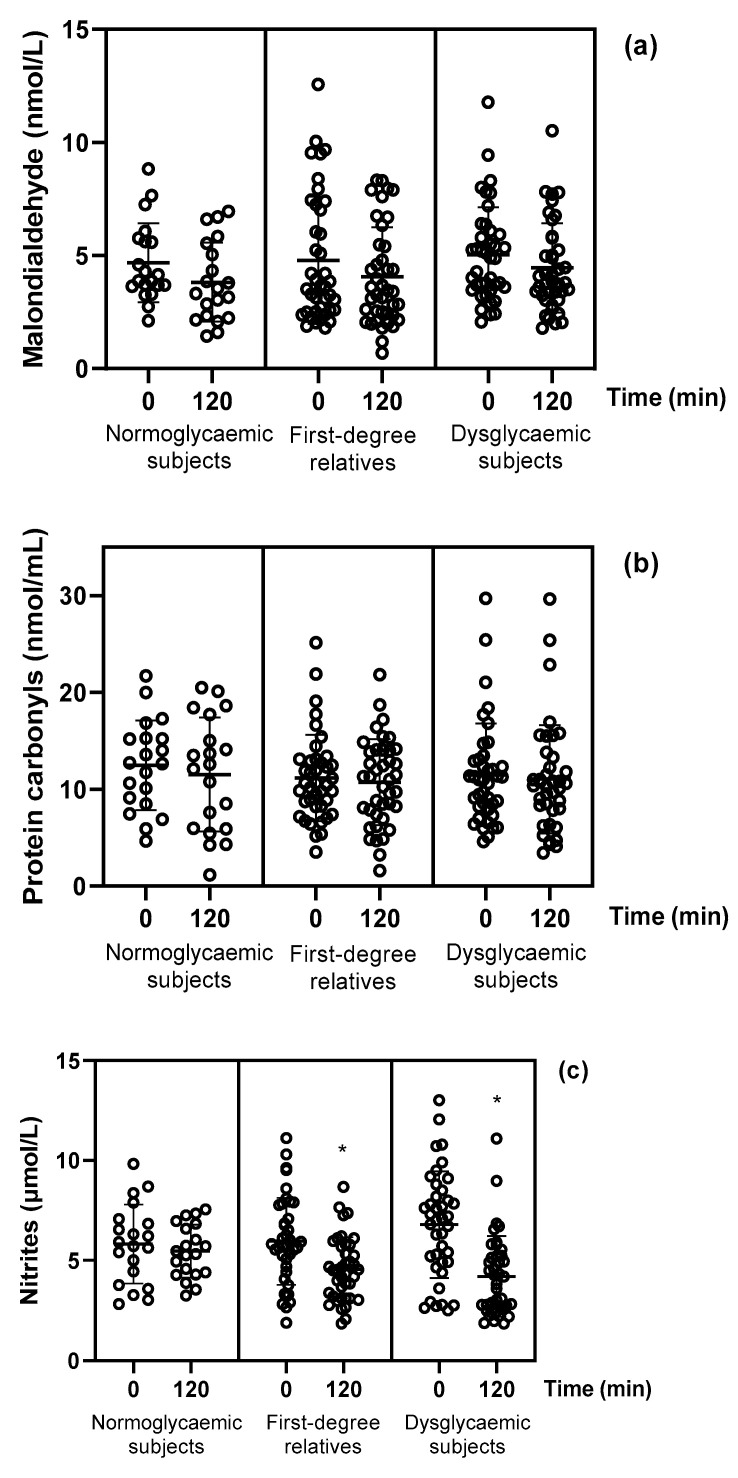
Changes in the plasma concentrations of (**a**) malondialdehyde, (**b**) protein carbonyls, (**c**) nitrites, (**d**) nitrates and (**e**) sum of nitrite and nitrate levels in the three study groups after 120 min glucose loading. * *p* < 0.05 for comparisons of 120 min vs. 0 min.

**Table 1 jcm-11-03299-t001:** Clinical and metabolic characteristics of the three study groups.

	Normoglycaemic Subjects (*n* = 20)	First-Degree Relatives (*n* = 40)	Dysglycaemic Subjects (*n* = 40)	*p*-Value
Age, years	37 ± 8	39 ± 7	43 ± 8	0.410
Sex (male/female), *n* (%)	12/8 (60/40)	22/18 (55/45)	22/18 (55/45)	0.833
Systolic BP (mmHg)	124 ± 14	127 ± 11	130 ± 9	0.116
Diastolic BP (mmHg)	77 ± 9	79 ± 8	83 ± 6	0.107
Risk factors, *n* (%)				
Hypertension	3 (15)	6 (15)	6 (15)	0.260
Dyslipidaemia	8 (40)	16 (40)	17 (43)	0.847
Current smoking	8 (40)	17 (43)	16 (40)	0.893
Family history CAD	4 (20)	7 (18)	8 (20)	0.972
Metabolic characteristics				
BMI (kg/m^2^)	28 ± 4	29 ± 5	30 ± 4 ***	0.834
Waist (cm)	100 ± 13	101 ± 14	102 ± 12	0.259
Hips (cm)	101 ± 10	104 ± 12	106 ± 9	0.201
Total cholesterol (mg/dL)	185 ± 15 *	214 ± 30 **	234 ± 26 ^‡‡‡^	<0.001
HDL cholesterol (mg/dL)	58 ± 8 ^‡^	50 ± 8 **	46 ± 7 ^‡‡‡^	<0.001
LDL cholesterol (mg/dL)	110 ± 15 *	136 ± 23 ^‡‡^	155 ± 24 ^‡‡‡^	<0.001
Triglycerides (mg/dL)	97 ± 22 ^‡^	136 ± 25 **	158 ± 28 ^‡‡‡^	<0.001
Fasting glucose (mg/dL)	91 ± 11	95 ± 7 ^‡‡^	115 ± 26 ^‡‡‡^	<0.001
Glucose at 120 min (mg/dL)	98 ± 15 *	106 ± 16 ^‡‡^	200 ± 50 ^‡‡‡^	<0.001
Fasting insulin (μU/mL)	8 ± 3 ^‡^	16 ± 8	15 ± 12 ***	0.034
Insulin at 120 min (μU/mL)	29 ± 13 ^‡^	58 ± 33 **	79 ± 70 ^††^	0.001
Matsuda index	5.5 ± 1.5 ^‡^	3 ± 1.2	2.9 ± 1.7 ^‡‡‡^	<0.001
ISI	94.3 ± 17.1 ^†^	75.2 ± 19.6 ^‡‡^	39 ± 13.4 ^‡‡‡^	<0.001

Data are given as mean ± standard deviation. BP, blood pressure; CAD, coronary artery disease; BMI, body mass index; HDL, high-density lipoprotein; LDL, low-density lipoprotein; ISI, insulin sensitivity index. *p*-value, *p* of model of the ANOVA or contingency table for comparisons between study groups. * *p* < 0.05, ^†^ *p* = 0.001, ^‡^ *p* < 0.001 for comparisons of normoglycaemic subjects vs. first-degree relatives. ** *p* < 0.05, ^‡‡^ *p* < 0.001 for first-degree relatives vs. dysglycaemic subjects. *** *p* < 0.05, ^††^ *p* = 0.001, ^‡‡‡^ *p* < 0.001 for dysglycaemic vs. normoglycaemic subjects by post hoc analysis with Bonferroni correction.

**Table 2 jcm-11-03299-t002:** Changes in the plasma concentrations of oxidative stress markers and cumulative nitrite and nitrate levels, endothelial glycocalyx integrity, coronary microcirculation and left ventricular myocardial function after 120 min glucose loading.

	Normoglycaemic Subjects (*n* = 20)	First-Degree Relatives (*n* = 40)	Dysglycaemic Subjects (*n* = 40)
	Malondialdehyde (nmol/L)		
0 min	4.68 ± 1.9	4.79 ± 2.8	5.03 ± 2.1
120 min	3.82 ± 1.7	4.07 ± 2.1	4.45 ± 2
Δ%	−18	−15	−11
	Protein carbonyls (nmol/mL)		
0 min	12.46 ± 4.9	11.15 ± 4.7	11.47 ± 5.3
120 min	11.51 ± 5.8	10.71 ± 4.4	11.02 ± 5.6
Δ%	−7	−4	−4
	Nitrites (μmol/L)		
0 min	5.82 ± 1.9	5.96 ± 2.1	6.7 ± 2.6
120 min	5.46 ± 1.3	4.57 ± 1.6 *	4.2 ± 2 *
Δ%	−6	−23	−37
	Nitrates (μmol/L)		
0 min	41.91 ± 16.3	44.39 ± 23.3	45.29 ± 18.9
120 min	40.67 ± 16.5	33.9 ± 20.6 *	32.36 ± 14.2 *
Δ%	−3	−24	−29
	Nitrites + Nitrates (μmol/L)		
0 min	47.73 ± 15.9	50.35 ± 23.1	52.08 ± 19.7
120 min	46.13 ± 16.4	38.47 ± 20.5 *	36.57 ± 14.4 *
Δ%	−3	−24	−30
	PBR 20–25 μm		
0 min	2.41 ± 0.3	2.5 ± 0.4 ^#^	2.52 ± 0.6 ^#^
120 min	2.34 ± 0.3	2.67 ± 0.4 *	2.77 ± 0.4 *
Δ%	−3	+7	+10
	CFR		
0 min	3.17 ± 0.57	3.15 ± 0.4	2.79 ± 0.35 ^#^
120 min	2.98 ± 0.56	2.81 ± 0.41 ^‡^	2.41 ± 0.31 ^‡^
Δ%	−6	−11	−14
	LV GLS (%)		
0 min	−19.2 ± 2.1	−18.4 ± 2.6 ^#^	−16.8 ± 2 ^#^
120 min	−19.2 ± 2.4	−17.6 ± 2.3 ^†^	−16.2 ± 1.4 ^‡^
Δ%	+0.1	−4	−4

Data are presented as mean ± standard deviation. PBR 20–25 μm, perfused boundary region of the sublingual microvessels ranged from 20 to 25 μm; CFR, coronary flow reserve; LV GLS, left ventricular global longitudinal strain. * *p* < 0.05, ^†^ *p* < 0.01, ^‡^ *p* < 0.001 for comparisons of 120 min vs. 0 min by ANOVA using post hoc analysis with Bonferroni correction. ^#^ *p* < 0.05 for first-degree relatives and dysglycaemic subjects vs. normoglycaemic subjects at baseline.

## Data Availability

The datasets generated and/or analyzed during the current study are not publicly available due to information that could compromise the privacy of research participants but are available from the corresponding author upon reasonable request. All other data are contained within the article.

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
