# Peer review of "Endothelial Dysfunction Is Associated with Decreased Nitric Oxide Bioavailability in Dysglycaemic Subjects and First-Degree Relatives of Type 2 Diabetic Patients"

_jcm, 2022, doi:10.3390/jcm11123299_

Round 1

Reviewer 1 Report

Here, Ikonomidis and colleagues investigated oxidative stress and nitrites/nitrates concentrations at baseline and during postprandial hyperglycemia in 40 first-degree relatives (FDR) of diabetic patients with normal oral glucose tolerance test (OGTT), 40 subjects with abnormal OGTT (dysglycaemic), and 20 subjects with normal OGTT (normoglycaemic). To that end, the authors assessed malondialdehyde (MDA), protein carbonyls (PC), nitrites/nitrates plasma levels, perfused boundary region (PBR-Glycocheck) of the sublingual microvessels, a marker of glycocalyx integrity, coronary flow reserve (CFR), and left ventricular global longitudinal strain (GLS) at 0 and 120 min during the OGTT. Interestingely, the authors found that compared with normoglycaemic subjects, FDR and dysglycaemic subjects had significantly decreased nitrites/nitrates levels (-3% versus -24% versus -30%, respectively), increased PBR and reduced CFR and GLS at 120min (p<0.05). The percent reduction of nitrites/nitrates was associated with abnormal Matsuda and ISI, reversely related with the percent increase of PBR (r=-0.60) and positively related with the percent decrease of CFR (r=0.39) and GLS (r=0.48) (p<0.05). The authors conclude that insulin resistance determines acute endothelial responses during OGTT leading to reduced nitric oxide production in FDR and dysglycaemic subjects. Albeit interesting, some issues remain that should be addressed by the authors:

Major:

- Causal inference is not appropriate just by the design of the study. The authors must refrain from infering causality by noting ‘x determines/causes y’. This must be revised throughout (this starts already at the level of the title, which implies that endothelial dysunfction is caused by decreased nitric oxide bioavailability; only observational evidence is provided, hence such conclusions are clearly overstated and not appropriate);

- This also applies to the main text where many conclusions are overstated and should be toned down;

- A flowchart depicting study design including time points at which measurements were performed should be included.

Author Response

Response to Reviewer 1 Comments

Point 1: Causal inference is not appropriate just by the design of the study. The authors must refrain from infering causality by noting ‘x determines/causes y’. This must be revised throughout (this starts already at the level of the title, which implies that endothelial dysunfction is caused by decreased nitric oxide bioavailability; only observational evidence is provided, hence such conclusions are clearly overstated and not appropriate);

Response 1: We acknowledge the reviewer’s comment. Following the reviewer’s suggestion, we have revised throughout the text. Also, we modified the title as follows:

“Endothelial dysfunction is associated with decreased nitric oxide bioavailability in dysglycaemic subjects and first-degree relatives of type 2 diabetic patients”

Point 2: This also applies to the main text where many conclusions are overstated and should be toned down;

Response 2: We acknowledge the reviewer’s comment. Following the reviewer’s suggestion, we have revised throughout the main text.

Point 3: A flowchart depicting study design including time points at which measurements were performed should be included.

Response 3: Following the reviewer’s suggestion, we have now added Figure 1 (revised manuscript, page 3) depicting study design and the time points at which measurements of the examined markers were performed.

Reviewer 2 Report

The study aimed to investigate oxidative stress and nitrites/nitrates concentrations at baseline and during postprandial hyperglycemia in first-degree relatives (FDR) of diabetic patients with normal OGTT, subjects with abnormal OGTT and subjects with normal OGTT. Although the topic was interesting, the very tiny sample size of the study was too hard to get a conclusion. Additionally, the conclusion has no novelty, for the relationship between insulin resistance, oxidative stress and endothelial dysfunction has been investigated before.

Author Response

Response to Reviewer 2 Comments

The study aimed to investigate oxidative stress and nitrites/nitrates concentrations at baseline and during postprandial hyperglycemia in first-degree relatives (FDR) of diabetic patients with normal OGTT, subjects with abnormal OGTT and subjects with normal OGTT. Although the topic was interesting, the very tiny sample size of the study was too hard to get a conclusion. Additionally, the conclusion has no novelty, for the relationship between insulin resistance, oxidative stress and endothelial dysfunction has been investigated before.

Response: We acknowledge the reviewer’s comments about the small sample size and the lack of novelty in the conclusion of the present study.

The small number of participants is now commented in the Study limitations section of the revised version as follows:

Page 12, line 419-21: “In addition, the study population was relatively young, with a mean age of 43 years. This, together with the small sample size of the study, prevents the results from being extended to an older population.”

In addition, we have added in the Conclusion section of the revised manuscript the following sentence:

Page 12, line 426-9: “It is the first time to the best our knowledge that decreased nitric oxide bioavailability have been linked to impaired endothelial glycocalyx integrity, coronary microcirculation and left ventricular myocardial function.”

Reviewer 3 Report

This manuscript is quite interesting. However, this reviewer raises some issues that should be addressed.

1- One of the groups studied is made up of 40 subjects with abnormal OGTT that the authors define as dysglycemic. They are subjects with plasma glucose levels >140 mg/dL in the 2-hour OGTT sample. However, patients with prediabetes or diabetes are not distinct from the authors. Certainly, some of these are diabetic since fasting glucose is 115±26 mg/dL, and 120 min glucose is 200±50 mg/dL. The authors should clarify the difference in nitric oxide production and endothelial dysfunction between diabetics and prediabetics.

2- The population studied is relatively young, with a mean age of around 40 years. This, together with the simple low size, prevents the results from being extended to an older population. These issues should be addressed in discussion as well as added in the limitations of the study.

3- Intriguingly, there is a significant relationship between the increase in TNF-alpha levels and the decrease in the endothelial function score in subjects with insulin resistance (Nutr Metab Cardiovasc Dis. 2007 May;17(4):274-9. doi: 10.1016/j.numecd.2005.11.014.). This issue with the above reference should be addressed in the discussion.

4- Recently some interesting reviews have explained the potential endothelial protective role of drugs such as metformin (Biomedicines. 2020 Dec 22;9(1):3. doi: 10.3390/biomedicines9010003.) and SGLT2i (Biomedicines. 2021 Sep 29;9(10):1356. doi: 10.3390/biomedicines9101356.) which are now also used in non-diabetic patients. This important issue should be commented on with the above references being discussed.

Author Response

Response to Reviewer 3 Comments

Point 1: One of the groups studied is made up of 40 subjects with abnormal OGTT that the authors define as dysglycemic. They are subjects with plasma glucose levels >140 mg/dL in the 2-hour OGTT sample. However, patients with prediabetes or diabetes are not distinct from the authors. Certainly, some of these are diabetic since fasting glucose is 115±26 mg/dL, and 120 min glucose is 200±50 mg/dL. The authors should clarify the difference in nitric oxide production and endothelial dysfunction between diabetics and prediabetics.

Response 1: We apologize for the lack of clarification regarding the differences in nitric oxide production and endothelial dysfunction between diabetic and prediabetic subjects.

Following the reviewer’s request, we have now providing detailed information on the differences between diabetic and prediabetic individuals in the revised version as follows:

Page 3, line 117-21: “In contrast, all dysglycaemic subjects (n = 40) showed glucose levels > 140 mg/dL at the 2-hour sample of the OGTT. Of those, 23 subjects had glucose levels at 120 min between 140-199 mg/dL and were defined as having impaired glucose tolerance (prediabetic subjects) whereas the remaining 17 subjects showed glucose levels at 120 min of the OGTT > 200 mg/dL and were defined as diabetic patients [13].”

Page 7, line 265-71: “In the group of dysglycaemic subjects, there were no significant differences in malondialdehyde, protein carbonyls and nitrites/nitrates levels between prediabetic (n = 23) and diabetic subjects (n = 17) at baseline (p > 0.05, data not shown). However, diabetic patients presented a greater reduction of nitrites, nitrates and of the sum of nitrites+nitrates (-42% versus -35%, p = 0.038, -32% versus -27%, p = 0.042, -34% versus -28%, p = 0.035, respectively) compared with prediabetic subjects whereas no significant changes were observed in malondiadehyde and protein carbonyls in both groups post-OGTT.”

Page 10, line 305-12: “Among dysglycaemic subjects (n = 40), diabetic patients (n = 17; 120 min glucose levels > 200 mg/dL) had similar values of LV GLS, higher values of PBR (2.78 ± 0.5 μm versus 2.42 ± 0.6 μm, p = 0.025) and more impaired CFR (2.66 ± 0.31 versus 3.02 ± 0.38, p = 0.043) compared with prediabetic subjects (n = 23; 120 min glycose levels between 140 - 199 mg/dL) at baseline. Post-OGTT, diabetic patients showed a greater increase of PBR (+13% versus +8%, p = 0.012) and a greater reduction of CFR (-18% versus -12%, p=0.039) whereas the change of LV GLS at 120 min did not differ between the two groups (diabetic and prediabetic subjects; p > 0.05).”

Point 2: The population studied is relatively young, with a mean age of around 40 years. This, together with the simple low size, prevents the results from being extended to an older population. These issues should be addressed in discussion as well as added in the limitations of the study.

Response 2: We acknowledge the reviewer’s comment. Following the reviewer’s suggestion, we have added in the Discussion section the following sentence:

Page 11, line 379-80: “Of note, the age of the subjects as well as the sample size of the present study are similar to those of the aforementioned studies.”

Also, we have added in the Study Limitations the following sentences:

Page 12, line 419-21: “In addition, the study population was relatively young, with a mean age of 43 years. This, together with the small sample size of the study, prevents the results from being extended to an older population.”

Point 3: Intriguingly, there is a significant relationship between the increase in TNF-alpha levels and the decrease in the endothelial function score in subjects with insulin resistance (Nutr Metab Cardiovasc Dis. 2007 May;17(4):274-9. doi: 10.1016/j.numecd.2005.11.014.). This issue with the above reference should be addressed in the discussion.

Response 3: Following the reviewer’s suggestion, we have now added in the Discussion section of the revised version the following sentence:

Page 11, line 392-5: “Intriguingly, Esposito et al. showed that tumor necrosis factor-α levels are elevated in individuals with insulin resistance and a high-fat meal induces further increase in its circulating levels in association with reduced endothelial function as assessed by the L-arginine test [31]”.

New reference: 31

Point 4: Recently some interesting reviews have explained the potential endothelial protective role of drugs such as metformin (Biomedicines. 2020 Dec 22;9(1):3. doi: 10.3390/biomedicines9010003.) and SGLT2i (Biomedicines. 2021 Sep 29;9(10):1356. doi: 10.3390/biomedicines9101356.) which are now also used in non-diabetic patients. This important issue should be commented on with the above references being discussed.

Response 4: Following the reviewer’s suggestion, we have now added in the Discussion section of the revised version the following sentence:

Page 11, line 400-4: “Interestingly, accumulating preclinical and clinical evidence support the potential endothelial protective role of anti-hyperglycemic agents such as metformin as well as gliflozins, which are now used in non-diabetic patients, by attenuating oxidative stress and endothelial inflammation [33,34].”

New references: 33 and 34

Round 2

Reviewer 1 Report

The authors have sufficiently addressed all the concerns raised.

Reviewer 2 Report

The authors are not willing to revise the manuscript according to the suggestion.

Reviewer 3 Report

No further comments.